# CLIP-Flow: Contrastive Learning with Iterative Pseudo labeling for Optical Flow

## Abstract

Synthetic datasets are often used to pretrain end-to-end optical flow networks, due to the lack of a large amount of labeled real scene data. But major drops in accuracy occur when moving from synthetic to real scenes. How do we better transfer the knowledge learned from synthetic to real domains? To this end, we propose CLIP-Flow, a semi-supervised iterative pseudo labeling framework to transfer the pretraining knowledge to the target real domain. We leverage large-scale, unlabeled real data to facilitate transfer learning with the supervision of iteratively updated pseudo ground truth labels, bridging the domain gap between the synthetic and the real. In addition, we propose a contrastive flow loss on reference features and the warped features by pseudo ground truth flows, to further boost the accurate matching and dampen the mismatching due to motion, occlusion, or noisy pseudo labels. We adopt RAFT as backbone and obtain an F1-all error of 4.11%, $i.e.$ a 19% error reduction from RAFT (5.10%) and ranking $2^{nd}$ place at submission on KITTI 2015 benchmark. Our framework can also be extended to other models, $e.g.$ CRAFT, reducing the F1-all error from 4.79% to 4.66% on KITTI 2015 benchmark.

## 1 Introduction

Optical flow is critical in many high level vision problems, such as action recognition (Simonyan & Zisserman, 2014; Sevilla-Lara et al., 2018; Sun et al., 2018b), video segmentation (Yang et al., 2021; Yang & Ramanan, 2021) and editing (Bonneel et al., 2015), autonomous driving (Janai et al., 2020) and so on. Traditional methods (Horn & Schunck, 1981; Menze et al., 2015; Ranftl et al., 2014; Zach et al., 2007) mainly focus on formulating flow estimation as solving optimization problems using hand-crafted features. The optimization is searched over the space of dense displacement fields between a pair of input images, which is often time-consuming. Recently, data driven deep learning methods Dosovitskiy et al. (2015); Ilg et al. (2017); Teed & Deng (2020) have been proved successful in estimating the optical flow thanks to the availability of all kinds of high quality synthetic datasets (Butler et al., 2012b; Dosovitskiy et al., 2015; Mayer et al., 2016; Krispin et al., 2016).

Most of the recent works (Dosovitskiy et al., 2015; Ilg et al., 2017; Teed & Deng, 2020; Jeong et al., 2022) mainly train on the synthetic datasets given that there is no sufficient real labeled optical flow datasets to be used to train a deep learning model. State-of-the-art (SOTA) models always get more accurate results on the synthetic dataset like Sintel (Butler et al., 2012a) than the real scene dataset like KITTI 2015 (Menze & Geiger, 2015). This is mainly because that the model tends to overfit the small training data, which echos in Tab. 1, $i.e.$, there is a big gap between the training F1-all error and test F1-all error when train and test on the KITTI dataset in all of the previous SOTA methods. Therefore, we argue that this gap in performance is because of dearth of real training data, and a big distribution gap between the synthetic data and real scene data. Although the model can perfectly explain all kinds of synthetic data, however, when dealing with real data, it performs rather unsatisfactorily. Our proposed work focuses on bridging the glaring performance gap between the synthetic data and the real scene data. As in previous data driven approaches, smarter and longer training strategies prove to be beneficial and helps in obtaining better optical flow results. Through our work we also try to find answers of the following two questions: (i) How to take advantage of the current SOTA optical flow models to further consolidate gain on real datasets? and (ii) How can we use semi-supervised learning along with contrastive feature representation learning strategies to effectively utilize the huge amount of unlabeled real data at our disposal?

Unsupervised visual representation learning (He et al., 2020; Chen et al., 2020b) has proved successful in boosting most of major vision related tasks like image classification, object detection and semantic segmentation to name a few. Work such as (He et al., 2020; Chen et al., 2020b) also emphasizes the importance of the contrastive loss, when dealing with huge dense dataset. Given that optical flow tasks generally lack real ground truth labels, we ask if leveraging the unsupervised visual representation learning boosts optical flow performance? In order to answer this question, we examine the impact of contrastive learning and pseudo labeling during training under a semi-supervised setting. We particularly conduct exhaustive experiments using KITTI-Raw Geiger et al. (2013) and KITTI 2015 (Menze & Geiger, 2015) datasets to evaluate its performance gain, and show encouraging results. We believe that gain seen in model's performance is reflective of the fact that employing representation learning techniques such as contrasting learning helps in achieving a much more refined 4D cost correlation volume. To constrain the flow per pixel, we employ a simple positional encoding of 2D cartesian coordinates on the input frames as suggested in (Liu et al., 2018), which further consolidates the gain achieved by contrastive learning. We follow this up with an iterative-flow-refinement training using pseudo labeling which further consolidates on the previous gains to give us SOTA results. At this point we would also like to highlight that we follow a specific and well calibrated training strategy to fully exploit the gains of our method. Without loss of generality, we use RAFT (Teed & Deng, 2020) as the backbone network for our experiments. To fairly compare our method with existing SOTA methods, we tested our proposed method on the KITTI 2015 test dataset, and we achieve the best F1-all error score among all the published methods by a significant margin.

To summarize our main contributions: 1) We provide a detailed training strategy, which uses SSL methods on top of the well known RAFT model to improve SOTA performance for optical flow estimation. 2) We present the ways to employ *contrastive learning* and *pseudo labeling* effectively and intelligently, such that both jointly help in improving upon existing benchmarks. 3) We discuss the positive impact of a simple 2D positional encoding, which benefits flow training both for Sintel and KITTI 2015 datasets.

## 2 RELATED WORK

**Optical flow estimation.** Maximizing visual similarity between neighboring frames by formulating the problem as an energy minimization (Black & Anandan, 1993; Bruhn et al., 2005; Sun et al., 2014) has been the primary approach for optical flow estimation. Previous works such as (Dosovitskiy et al., 2015; Ilg et al., 2017; Ranjan & Black, 2017; Sun et al., 2018a; 2019; Hui et al., 2018; 2020; Zou et al., 2018) have successfully established efficacy of deep neural networks in estimating optical flow both under supervised and self-supervised settings. Iterative improvement in model architecture and better regularization terms has been primarily responsible for achieving better results. But, most of these works fail to better handle occlusion, small fast-moving objects, capture global motion and rectify and recover from early mistakes.

To overcome these limitations, Teed & Deng (2020) proposed RAFT, which adopts a learning-to-optimize strategy using a recurrent GRU-based decoder to iteratively update a flow field f which is initialized at zero. Inspired by the success of RAFT, there has been a number of variants such as CRAFT (Sui et al., 2022), GMA (Jiang et al., 2021), Sparse volume RAFT (Jiang et al., 2021) and FlowFormer (Huang et al., 2022), all of which benefits from all-pair correlation volume way of estimating optical flow. The current state-of-the-art work RAFT-OCTC (Jeong et al., 2022) also uses RAFT based architecture, and it imposes consistency based on various proxy tasks to improve flow estimation. Considering RAFT's effectiveness, generalizability and relatively smaller model size, we adopt RAFT (Teed & Deng, 2020) as our base architecture and employ semi-supervised iterative pseudo labeling together with the contrastive flow loss, to achieve a state-of-the-arts result on KITTI 2015 (Menze & Geiger, 2015) benchmark.

**Semi-Supervised and Representation Learning.** Semi-supervised learning (SSL) and representation learning have shown success for a range of computer vision tasks, both during the pretext task training and during specific downstream tasks. Most of these methods leverage contrastive learning (Chen et al., 2020a;b; He et al., 2020), clustering (Caron et al., 2020) and pseudo-labeling (Caron et al., 2021; Chen & He, 2021; Grill et al., 2020; Hoyer et al., 2021) as an enforcing mechanism. Recent works such as (Caron et al., 2020; 2021; Chen et al., 2020a;b; 2021; Grill et al., 2020; He

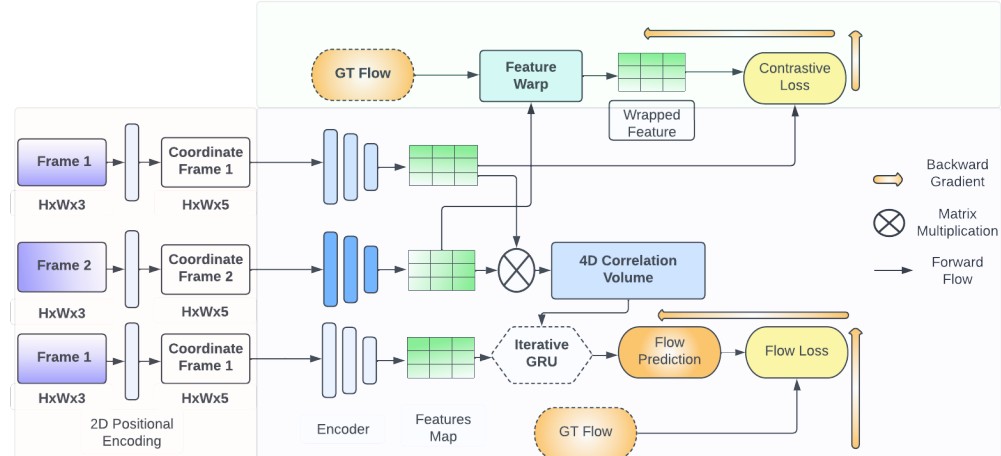

Figure 1: Overview of our Contrastive Flow Model(CLIP-Flow). Input frames are firstly positionally encoded, according to individual 2D pixel locations. Subsequently *contrastive loss* are applied on encoded features to enforce pixel wise consistency using GT Flow.

et al., 2020; Xie et al., 2021b; Yun et al., 2022) have empirically shown benefits of SSL for downstream tasks such as image classification, object detection, instance and semantic segmentation. These works leverage contrastive loss in different shapes and forms to facilitate better representation learning. For example, (He et al., 2020; Chen et al., 2020b) advocate for finding positive and negative keys with respect to a given encoded query to enforce a contrastive loss. On similar lines, for dense prediction tasks, studies such as (O Pinheiro et al., 2020; Xiao et al., 2021; Xie et al., 2021a) enforce matching overlapping regions between two augmented images, where as (Yun et al., 2022) looks to form a positive/negative pair between adjacent patches. As part of our approach we use contrastive flow loss between features of the neighboring frames, where we draw a one-to-one positive pair relations between the reference features and the warped features using (pseudo) ground truth flow or flow estimates, as shown in Fig. 1.

Pseudo labeling is another important approach in SSL training paradigm. Some studies leverage pseudo labeling to generate training labels as part of consistency training (Yun et al., 2019; Olsson et al., 2021; Hoyer et al., 2021), while other works propose to use pseudo labeling to improve training pretext task training as in (Caron et al., 2021; Chen & He, 2021; Grill et al., 2020). Rather, we use pseudo labeling for an *iterative refinement* mechanism, through which we effectively distill the correct flow estimate for the KITTI-Raw dataset (Geiger et al., 2013). To the best of our knowledge, ours is the first work which leverages both contrastive learning and pseudo labeling for estimating optical flow in an SSL fashion.

## 3 APPROACH

In this section, we describe our method CLIP-Flow, a semi-supervised framework for optical flow estimation by iterative pseudo labeling and contrastive flow loss. Based on two SOTA optical flow networks RAFT (Teed & Deng, 2020) and CRAFT (Sui et al., 2022) as backbone (*c.f.* Sec.3.1), we obtain non-trivial improvement by leveraging our iterative pseudo labeling (PL) (*c.f.* Sec. 3.2) and the proposed contrastive flow loss (*c.f.* Sec. 3.3). It should be noted that our CLIP-Flow can be easily extended to other optical flow networks, *e.g.* FlowNet (Dosovitskiy et al., 2015; Ilg et al., 2017), SpyNet (Ranjan & Black, 2017) and PWC-Net (Sun et al., 2018a), with little modification.

### 3.1 PRELIMINARIES

Given two consecutive RGB images $I_1, I_2 \in \mathbb{R}^{H \times W \times 3}$, the optical flow $\mathbf{f} \in \mathbb{R}^{H \times W \times 2}$ is defined as a dense 2D motion field $\mathbf{f} = (f_u, f_v)$, which maps each pixel $(u, v)$ in $I_1$ to its counterpart $(u', v')$ in $I_2$, with $u' = u + f_u$ and $v' = v + f_v$.

**RAFT.** Among end-to-end optical flow methods (Dosovitskiy et al., 2015; Ilg et al., 2017; Ranjan & Black, 2017; Sun et al., 2018a; Sui et al., 2022; Jeong et al., 2022; Teed & Deng, 2020), RAFT

(Teed & Deng, 2020) features a learning-to-optimize strategy using a recurrent GRU-based decoder to iteratively update a flow field $\mathbf{f}$ which is initialized at zero. Specifically, it extracts features using a convolutional encoder $g_\theta$ from the input images $I_1$ and $I_2$, and outputs features at 1/8 resolution, *i.e.*, $g_\theta(I_1) \in \mathbb{R}^{H' \times W' \times C}$ and $g_\theta(I_2) \in \mathbb{R}^{H' \times W' \times C}$, where $H'{=}H/8$ and $W'{=}W/8$ for spatial dimension and $C{=}256$ for feature dimension. Also, a context network $h_\theta$ is applied to the first input image $I_1$. Then all-pair visual similarity is computed by constructing a 4D correlation volume $\mathbf{V} \in \mathbb{R}^{H' \times W' \times H' \times W'}$ between features $g_\theta(I_1)$ and $g_\theta(I_2)$. It can be computed via matrix multiplication as $\mathbf{V} = g_\theta(I_1) \cdot g_\theta^T(I_2)$, *i.e.*, $\left( \mathbb{R}^{(H' \cdot W') \times C}, \mathbb{R}^{C \times (H' \cdot W')} \right) \mapsto \mathbb{R}^{(H' \cdot W') \times (H' \cdot W')}$, which is further reshaped to $\mathbf{V} \in \mathbb{R}^{H' \times W' \times H' \times W'}$. Then RAFT builds a 4-layer coorelation pyramid $\{\mathbf{V}^s\}_{s=1}^4$ by pooling the last two dimensions of $\mathbf{V}$ with kernel sizes $2^{s-1}$, respectively. The GRU-based decoder estimates a sequence of flow estimates $\{\mathbf{f}_1, \ldots, \mathbf{f}_T\}$ ($T{=}12$ or $24$) from a zero initialized $\mathbf{f}_0 = \mathbf{0}$. RAFT attains high accuracy, strong generalization as well as high efficiency. We take RAFT as the backbone and achieve boosted performance, *i.e.* F1-all errors of 4.11 (ours) vs 5.10 (raft) on KITTI-2015 (Menze & Geiger, 2015) (*c.f.* Tab. 1).

**CRAFT.** To overcome the challenges of large displacements with motion blur and the limited field of view due to locality of convolutional features in RAFT, CRAFT (Sui et al., 2022) proposes to leverage transformer layers to learn global features by considering long-range dependence, and hence revitalize the 4D correlation volume $\mathbf{V}$ computation as in RAFT. We also use CRAFT as the backbone and attain improvement, *i.e.* F1-all errors of 4.66 (ours) vs 4.79 (craft) on KITTI-2015 (Menze & Geiger, 2015) (*c.f.* Tab. 1).

## 3.2 ITERATIVE PSEUDO LABELING

Deep learning based optical flow methods are usually pretrained on synthetic data [1] and fine-tuned on small real data. This begs an important question: *How to effectively transfer the knowledge learned from synthetic domain to real world scenarios and bridge the big gap between them?* Our semi-supervised framework is proposed to improve the performance on real datasets $\mathcal{D}_R$, by iteratively transferring the knowledge learned from synthetic data $\mathcal{D}_S$ and/or a few of available real datasets $\mathcal{D}_R^{tr}$ (with sparse or dense ground truth optical flow labels). Without loss of generality, we assume that the real data $\mathcal{D}_R$ consists of i) a small amount of training data $\mathcal{D}_R^{tr}$ (*e.g.* KITTI 2015 (Menze & Geiger, 2015) training set with 200 image pairs) due to the expensive and tedious labeling by human, ii) a number of testing data $\mathcal{D}_R^{te}$ (*e.g.* KITTI 2015 test set with 200 pairs), and iii) a large amount of unlabeled data $\mathcal{D}_R^u$ (*e.g.* KITTI raw dataset (Geiger et al., 2013) having 84,642 images pairs) which is quite similar to the test domain. Therefore, we propose to use the unlabeled, real KITTI Raw data by generating pseudo ground truth labels using a master (or teacher) model to transfer the knowledge from pretraining on synthetic data or small real data to real data KITTI 2015 test set.

As shown in Fig. 2, our semi-supervised iterative pseudo labeling training strategy includes 3 steps: 1) Training on a large amount of unlabeled data ($\mathcal{D}_R^u$) supervised by a master (or teacher) model, which is chosen at the beginning as a model pretrained on large-scale synthetic and small real datasets, 2) Conducting $k$-fold cross validation on the labeled real dataset ($\mathcal{D}_R^{tr}$) to find best hyper-parameters, *e.g.* training steps for finetuing $S_{ft}$, and 3) finetuning our model on the labeled dataset ($\mathcal{D}_R^{tr}$) using the best hyper-parameters selected above, and updating the finetuned model as a new version of the master (or teacher) model to repeat those steps for next iteration, until the pre-defined iteration steps $N$ is reached or the gain of evaluation accuracy on test set ($\mathcal{D}_R^{te}$) is marginal. The detailed algorithm is illustrated in Alg. 1.

**Semi-supervised learning on unlabeled real dataset.** Our proposed iterative pseudo labeling method aims at dealing with real imagery that usually lacks ground truth labels and is difficult to be accurately modeled by simulators due to reflective surfaces, sensor noise and illumination conditions (Cai et al., 2020). As shown in Alg. 1 and Fig. 2-A, a pretrained baseline model $\phi_{bs}$, *e.g.* RAFT, and our pretrained model $\phi_{our}^0$, *e.g.* RAFT-CF, a variant of the baseline RAFT by adding our contrastive flow loss (*c.f.* Sec 3.3) are provided and they are pretrained using the same input datasets of synthetic images ($\mathcal{D}_S$, including C+T+S for FlyingChairs, FlyingThings3D, Sintel, respectively) and a small number of real ones ($\mathcal{D}_R^{tr}$, including K+H for KITTI 2015 and HD1K). Then based on the pretrained model $\phi_{our}^0$ (*c.f.* line 6 in Alg. 1), we train our model $\phi_{our}$ on the large unlabeled real dataset ($\mathcal{D}_R^u$)

---

[1]We assume the synthetic data is at large scale and have ground truth optical flow maps.

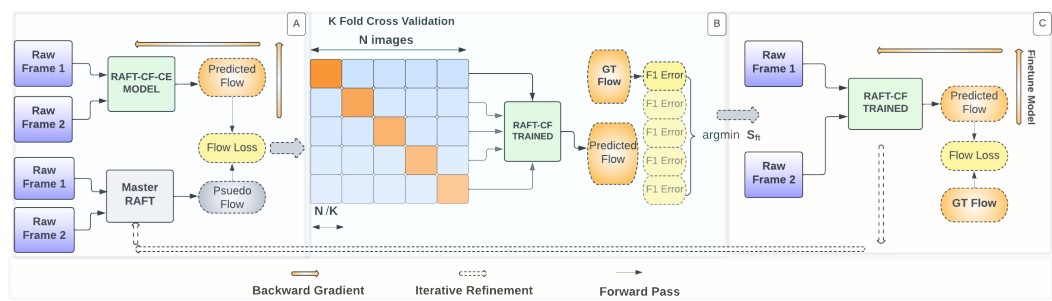

Figure 2: Overview of our semi-supervised iterative pseudo labeling training strategy, which can be easily applied to existing SOTA architectures. Taking RAFT (Teed & Deng, 2020) as an example, given baseline **RAFT** and **RAFT-CF-CE** (with our proposed Contrastive Flow loss and Coordinate encoding), both pretrained on C+T+S+K+H datasets, short for FlyingChairs, FlyingThings3D, Sintel, KITTI 2015 and HD1K, respectively. More details about training is provided in 4.2.

---

**Algorithm 1** Semi-supervised iterative pseudo labeling for optical flow estimation

---

**Input:** synthetic datasets $\mathcal{D}_S$, real datasets $\mathcal{D}_R$ (including training data $\mathcal{D}_R^{tr}$, test data $\mathcal{D}_R^{te}$ and a large amount of unlabeled data $\mathcal{D}_R^u$)

**Input:** $\phi_{bs}$         $\triangleright$ baseline (*e.g.* RAFT and CRAFT) pretrained on $\mathcal{D}_S \cup \mathcal{D}_R^{tr}$

**Input:** $\phi_{our}^0$         $\triangleright$ Ours ($\phi_{bs}$ + pseudo labeling + contrastive flow loss)

1:   $i \leftarrow 0$ and $N \leftarrow$ *e.g.* 3         $\triangleright$ index and total number of iterative pseudo labeling

2:   $s \leftarrow 0$ and $S \leftarrow$ *e.g.* $1.2 \times 10^6$         $\triangleright$ index and total training steps on $\mathcal{D}_R^u$

3:   **while** $i < N$ **do**         $\triangleright$ training on real data $\mathcal{D}_R$

4:      $s \leftarrow 0$

5:      $\hat{\mathbf{f}} = \phi_{bs}\left(\mathcal{D}_R^u\right)$ if $i = 0$ else $\phi_{our}\left(\mathcal{D}_R^u\right)$         $\triangleright$ pseudo ground truth

6:      $\phi_{our} \leftarrow \phi_{our}^0$         $\triangleright$ initialization with our model $\phi_{our}^0$ pretrained on $\mathcal{D}_S \cup \mathcal{D}_R^{tr}$

7:      **while** $s \leq S$ **do**         $\triangleright$ training on unlabeled real data $\mathcal{D}_R^u$

8:         $\phi_{our} \leftarrow \mathcal{L}\left(\phi_{our}\left(\mathcal{D}_R^u\right), \hat{\mathbf{f}}\right)$         $\triangleright$ training with pseudo label $\hat{\mathbf{f}}$ by loss function $\mathcal{L}$

9:         $s \leftarrow s + 1$

10:      **end while**

11:      $s \leftarrow 0$ and $S_{ft} \leftarrow$ k-fold$\left(\phi_{our}, \mathcal{D}_R^{tr}\right)$         $\triangleright$ find best steps $S_{ft}$ for finetuing by $k$-fold cross validation

12:      **while** $s \leq S_{ft}$ **do**

13:         $\phi_{our} \leftarrow \mathcal{L}\left(\phi_{our}\left(\mathcal{D}_R^{tr}\right), \hat{\mathbf{f}}\right)$         $\triangleright$ finetuning on real train data $\mathcal{D}_R^{tr}$ by loss $\mathcal{L}$

14:         $s \leftarrow s + 1$

15:      **end while**

16:      $i \leftarrow i + 1$         $\triangleright$ increment index $i$ for next pseudo labeling iteration

17: **end while**

---

with the supervision of the pseudo ground truth labels $\hat{\mathbf{f}}$ (*c.f.* line 5 in Alg. 1). The pseudo labels $\hat{\mathbf{f}}$ are generated by a master model (*c.f.* Fig. 2-A), and the master model is initialized as the pretrained baseline $\phi_{bs}$ at the first step, and will be updated in the following iteration with our new model $\phi_{our}$.

**k-Fold Cross validation on labeled small real dataset** Gathering annotated data for real world applications may be too expensive or infeasible. In order to take most of the labeled dataset, we use $k$-fold cross validation to find the best training hyper-parameters, especially, the best training steps $S_{ft}$ for finetuing on the whole real dataset ($\mathcal{D}_R^{tr}$, *e.g.* KITTI 2015 training set). As shown in the Fig. 2-B, we divide the validation dataset into $k$ folds (*e.g.* $k$=5), and we evaluate the trained model $\phi_{our}$ on the corresponding part of the validation set. As shown in Fig. 3, we pick the best hyper-parameter according to the lowest average validation error.

**Finetuning the model.** After getting the best hyper-parameterw via the $k$-fold cross validation, we use the obtained hyper parameters to finetune our model $\phi_{our}$ on the whole real labeled data ($\mathcal{D}_R^{tr}$ (*c.f.* line 13 in Alg. 1). We use this finefuned model $\phi_{our}$ to update the master (or teacher) model to produce second round pseudo ground truth optical flow labels. These steps will be repeated until the predefined iteration $N$ is reached or the gain of accuracy on real dataset ($\mathcal{D}_R$ is marginal).

### 3.3 CONTRASTIVE FLOW

**Self-supervised Contrastive Learning.** Contrastive learning first introduced by (Hadsell et al., 2006) has become increasingly successful for self-supervised representation learning (Wu et al., 2018; Oord et al., 2018; Hjelm et al., 2019; Bachman et al., 2019; Chen et al., 2020a; He et al., 2020; Chen et al., 2021) in computer vision. It learns visual representations such that two similar (positive) points have a small distance and two dissimilar (negative) points have a large distance. This can be formulated as a dictionary look-up problem given an encoded query $q$ and a set of encoded samples $\{k_0, k_1, k_2, \dots\}$ as the keys of a dictionary (He et al., 2020; Chen et al., 2021). A general formula of a contrastive loss function, called InfoNCE (Oord et al., 2018), is defined as:

$$\mathcal{L}_q = -\log \frac{\exp\left(q \cdot k^+/\tau\right)}{\exp\left(q \cdot k^+/\tau\right) + \sum_{k^-} \exp\left(q \cdot k^-/\tau\right)} \tag{1}$$

where $k^+$ is the positive sample of the query $q$, and the set $\{k^-\}$ consists of negative samples of $q$, $\tau$ is a temperature hyper-parameter, and the operator "$\cdot$" is dot product. In general, the query representation is $q = g_q(x^q)$ where $g_q$ is an encoder network and $x^q$ is an input query image, and similarly, $k = g_k(x^k)$ for keys.

**Semi-supervised Contrastive Flow.** To improve the representations for optical flow, we propose a contrastive flow loss to explicitly supervise the network training for learning better features and hence an informative correlation volume in RAFT (Teed & Deng, 2020). Given the input images $I_1$ and $I_2$, the extracted features $g_\theta(I_1)$ and $g_\theta(I_2)$, and the optical flow $\mathbf{f}$ [2], a pixel index $i$ in $I_1$ with the feature $g_i^1 \in g_\theta(I_1)$ is warped to a corresponding pixel index $j$ in $I_2$, with $j = i + \mathbf{f}_i$ in $I_2$ and the corresponding feature $g_j^2 \in g_\theta(I_2)$ is sampled from feature map $g_\theta(I_2)$ via bilinear interpolation. We consider the corresponding pair $(i, j)$ as a positive pair and other samples $(i, k)$ for $k \neq j$ as negative pairs. Therefore, we define the contrastive flow loss for index $i$ as

$$l_i = \frac{\exp\left(g_i^1 \cdot g_j^2/\tau\right)}{\exp\left(g_i^1 \cdot g_j^2/\tau\right) + \sum_{k \in I_2, k \neq j} \exp\left(g_i^1 \cdot g_k^2/\tau\right)} \tag{2}$$

And the contrastive flow loss over all the valid pixels which have (pseudo) ground truth in image $I_1$ is defined as

$$L_{CT} = \frac{1}{N_v} \sum_{i=0}^{N_v - 1} -\log l_i \tag{3}$$

where $N_v$ is the number of valid pixels and recall $j = i + \mathbf{f}_i$. The loss $L_{CT}$ in Eq. 3 can be efficiently computed as matrix multiplication.

**Coordinate Encoding.** Optical flow is the task of estimating per-pixel motion between video frames. Therefore, the coordinate of each pixel should be an important cue when predicting the optical flow for each pixel pair in $I_1$ and $I_2$. We proposed to concatenate 2D-coordinate map of each pixel as two additional channels to the original input frames. As shown in Fig. 1, the new input pair becomes $I_1', I_2' \in \mathbb{R}^{H \times W \times 5}$. In the ablation study (Sec. 4.3), we demonstrate that incorporating the coordinate encoding consistently improves the optical flow estimates.

## 4 EXPERIMENTS

### 4.1 EXPERIMENTAL SETTING AND DATASET

In our experiments, we have mainly focused on two recent and well-known base models, which are RAFT(Teed & Deng, 2020) and CRAFT(Sui et al., 2022), to verify the efficacy of our proposed training strategy. Our pretrained model based on RAFT and CRAFT are achieved by pretraining on dataset such as Sintel(Butler et al., 2012a), HD1K (Krispin et al., 2016), FlyingChair(Dosovitskiy et al., 2015), and FlyingThing(Mayer et al., 2016). We use similar training and hyperparamter

---

[2]It could be a ground truth optical flow or a pseudo label predicted by RAFT pretrained in synthetic datasets.

setting to achieve these models, as described in the original works. We refer readers to (Teed & Deng, 2020; Sui et al., 2022) for more details. Training during iterative refinement using pseudo labelling, is done on KITTI flow dataset(Menze & Geiger, 2015; Menze et al., 2018), where we use KITTI-Raw dataset, which consists of about 84642 images pairs but without any ground-truth labels for optical flow. We call the publicly available 200 KITTI flow dataset with manually labeled optical flow groundtruth as KITTI-Flow-Val, and the other 200 reserved KITTI flow testing data as KITTI-Flow-test.

For a fair analysis, we conduct exhaustive ablation experiments and use the standard KITTI-Flow-test dataset, for evaluating our models. In the next few subsections we discuss in detail the results achieved on KITTI dataset using our training strategy. We go on to show that our proposed model (RAFT-CF) along with iterative pseudo labelling helps us achieve state-of-art result for KITTI dataset and shows significant improvement for synthetic dataset as well.

## 4.2 RESULTS ON KITTI

**RAFT Model** Training on KITTI dataset could be broadly subdivided in to three stages as shown in the 2 and is discussed in section 3.2. All 3 stages defined in 2(A, B, and C), forms one iteration of pseudo-label enabled training. In the first stage (Fig.2-A), we train our proposed RAFT-CF-CE (RAFT with Contrastive flow with Coordinate embedding) on KITTI-Raw dataset using the pseudo flow labels, generated using original pretrained RAFT model as our Master RAFT model for 1.2m steps. In this stage our *candidate* RAFT-CF-CE model will be updated by the supervision of the pseudo flows. In the second stage (Fig.2-B), we take the *updated* RAFT-CF-CE model from part A and perform 5-fold cross validation on the KITTI-Flow-Val, by dividing the 200 image pairs into 5 parts, where each part includes 40 images pair. We conduct a total of 5 experi-

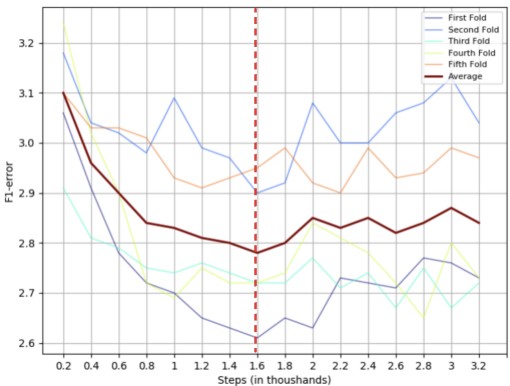

Figure 3: The F1-all error score curve at every 200 steps during the 5-fold cross validation. The (bold) brown curve is the average F1-all error across all the 5-fold validation experiments.

ments on the *updated* RAFT-CF-CE model, and for each group of cross-validation, we evaluate the F1-all error score at every 200 steps. We then calculate the average F1-all error score for each step through all the 5 groups of training, and pick the one that gives the lowest F1-all error score as the candidate best suited for fine-tuning our *updated* RAFT-CF-CE model. As seen in 3, we find that, we achieve an inflection point around 1.6k training steps, which gives us the best average accuracy on the validation dataset. Through this cross validation step, we efficiently narrow down the optimum number of training steps required to further fine-tune the model on KITTI-Flow-Val dataset. It also brings out the fact that, the model tends to grossly over-fit the small training data even after 2k steps.

Finally, In the third stage (Fig. 2-C), we finetune the *updated* RAFT-CF-CE-PL model from part A with the whole KITTI-Flow-Val for the optimum steps, which we get from Part B. After finetune, we get our final updated RAFT-CF-CE model for the current iteration. We repeat this process for iterative refinement for few more iterations till the model converges.

The comparative results on KITTI-Flow-Val and KITTI-Flow-test among all the published SOTA methods is presented in Tab. 1. Results seen in Tab. 1, further validates robustness of our approach, as even-though all of our model have higher training error, it performs much better on test dataset. Unlike other SOTA methods, which get lower training error but higher testing error, clearly reflecting towards an over-fitted model. This is also because of the fact that we follow a semi-supervised approach leveraging contrastive and pseudo labels during our training. Therefore, we claim that with the proposed training strategy, our model can be trained for more iterations and even more steps without any data-overfitting issues. In our case, we conducted 3 iterations of iterative training and the last iteration has 5 million steps, giving us a F1-all error score of **4.11** on the KITTI-Flow-test with **2.78** training error on KITTI-Flow-Val. Qualitative improvement using our model is presented

| Method | Training dataset | Sintel (train) | | KT15 (train) | | Sintel (test) | | KT15 (test) |
|---|---|---|---|---|---|---|---|---|
| | | Clean | Final | F1-epe | F1-all | Clean | Final | F1-all |
| LiteFLowNet2 (Hui et al., 2018) | C+T+S+K+H | (1.30) | (1.62) | (1.47) | (4.80) | 3.48 | 4.69 | 7.62 |
| PWC-Net+ (Sun et al., 2019) | | (1.71) | (2.34) | (1.50) | (5.30) | 3.45 | 4.60 | 7.72 |
| VCN (Yang & Ramanan, 2019) | | (1.66) | (2.24) | (1.16) | (4.10) | 2.81 | 4.40 | 6.30 |
| MaskFlowNet (Zhao et al., 2020) | | - | - | - | - | 2.52 | 4.17 | 6.10 |
| RAFT (Teed & Deng, 2020) | | (0.76) | (1.22) | (0.63) | (1.50) | 1.61* | 2.86* | 5.10 |
| GMA (Jiang et al., 2021) | | (0.62) | (1.06) | (0.57) | (1.20) | **1.39*** | 2.47* | 5.15 |
| RAFT-OCTC (Jeong et al., 2022) | | (0.73) | (1.23) | (0.67) | (1.70) | 1.82 | 3.09 | 4.72 |
| RAFT-OCTC† | | (0.74) | (1.24) | (0.71) | (2.00) | 1.58 | 2.95 | - |
| RAFT-OCTC‡ | | - | - | (0.78) | (2.30) | 1.41* | 2.57* | 4.33 |
| CRAFT (Sui et al., 2022) | | (0.60) | (1.06) | (0.58) | (1.34) | 1.45 | **2.42** | 4.79 |
| RAFT-A (Sun et al., 2021) | A+T+S+K+H | - | - | - | - | 2.01 | 3.14 | 4.78 |
| Ours (RAFT-CF-CE) | C+T+S+K+H | - | - | - | - | 1.52 | 2.65 | - |
| Ours (RAFT-CF-CE-PL1) | | - | - | (1.24) | (3.99) | - | - | 4.38 |
| Ours (RAFT-CF-CE-PL2) | | - | - | (1.10) | (3.18) | - | - | 4.13 |
| Ours (RAFT-CF-CE-PL3) | | - | - | (1.02) | (2.78) | - | - | **4.11** |
| Ours (CRAFT-CE-PL1) | | - | - | (1.67) | (5.12) | - | - | 4.66 |
| Ours (CRAFT-CF-CE-PL1) | | - | - | (1.71) | (5.23) | - | - | 4.68 |

Table 1: Optical flow results on Sintel and KITTI 2015. RAFT-CF-CE is our modified version of baseline RAFT, by adding contrastive flow (CF) loss and coordinate encoding (CE). Similarly, -PL is for the adding of our proposed semi-supervised iterative pseudo labeling (PL) training strategy, and -PL1, -PL2 and -PL3 represent the 1st, 2nd and 3rd iteration respectively. The similar setup is for the backbone CRAFT Sui et al. (2022). *Results evaluated with the "warm-start" strategy detailed in RAFT. (Result) denotes a result on training sets, listed here for reference only. Bold for best results and the second best results are underlined.

in Fig. 5 on KITTI -Flow-test 2015 (Menze & Geiger, 2015). The right column is our predicted optical flow results, the middle column is the results from the baseline model RAFT. Our model, as seen from the results, is make finer flow predictions and also better handles occluded pixels.

**CRAFT** In order to demonstrate generalizability of the proposed contrastive loss and iterative pseudo labeling training strategy, we also trained the most recently released state-of-art model CRAFT(Sui et al., 2022) with our training strategy. We follow a similar 3 stage approach as described in RAFT Model paragraph of 4.2 for the CRAFT model. During our experiments, we found that training CRAFT based model takes unexpectedly longer time compared to one based on RAFT. Due to this constrain, we ran the CRAFT based CRAFT-CE-PL and CRAFT-CF-CE-PL model for 1 iteration with 620k and 420k steps respectively. We then tested our final model on KITTI-Flow-test and we get F1-all error score of **4.66** and **4.68**, which outperforms original CRAFT model(4.79 F1-all error) by a good margin, as shown in table 1.

## 4.3 ABLATIONS

| iteration | steps | F1-all(val) | F1-all(test) |
|---|---|---|---|
| 1st | 1.2m | 3.99 | 4.38 |
| 2nd | 1.2m | 3.42 | - |
| 3rd | 1.2m | 3.18 | 4.13 |
| 3rd | 5m | 2.78 | 4.11 |

Table 2: The comparison of F1-all error on KITTI-Flow-Val and KITTI-Flow-test at different iteration and different training steps.

| Models (@step 1.2m) | F1-all (validation) | F1-all (test) |
|---|---|---|
| RAFT-PL | 4.531 | 4.48 |
| RAFT-CE-PL | 4.45 | - |
| RAFT-CF-CE-PL | 3.99 | 4.38 |

Table 3: The comparison of F1-all error KITTI-Flow-Val and KITTI-Flow-test for different RAFT based model.

As outlined in the earlier sections, our method can be broadly divided in to three independent subparts, which are: 1) Employing iterative training with **P**seudo **L**abeling (PL), 2) Adding **C**ontrastive **F**low loss to the baseline encoder features (CF), and 3) Applying **C**oordinate **E**ncoding (CE) to the raw input frames. Here we will analyze how each of them contribute the improved performance. 4-(b) shows the F1-all error score on KITTI-Flow-Val. The top red curve is the RAFT-PL model on trained on KITTI-Raw. The middle green curve is model RAFT-CE-PL, and the bottom blue is the RAFT-CF-CE-PL model. Through the experiments it shows that the proposed iterative pseudo labeling training strategy itself will highly increase the predicted optical flow accuracy. Adding coordinate embedding and contrastive flow loss will further increase the optical flow accuracy. The quantitative results shown in Table 3.

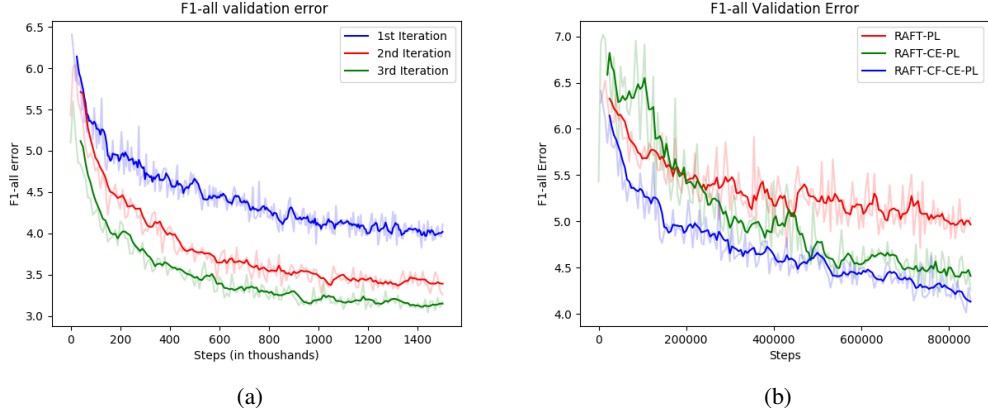

(a)                                                          (b)

Figure 4: (a) F1-all error curve on KITTI-Flow dataset when training (Our) RAFT-CF-CE-PL model on the KITTI-Raw dataset. 1st iteration (blue curve) is supervised by pseudo flow generated using the original RAFT. In 2nd and 3rd iteration, it is supervised by the updated RAFT-CF-CE-PL1 and RAFT-CF-CE-PL1 model respectively. (b) The curve shows, F1-all error on KITTI-Flow dataset for different RAFT based model. We see our RAFT-CF-CE-PL model, performs the best.

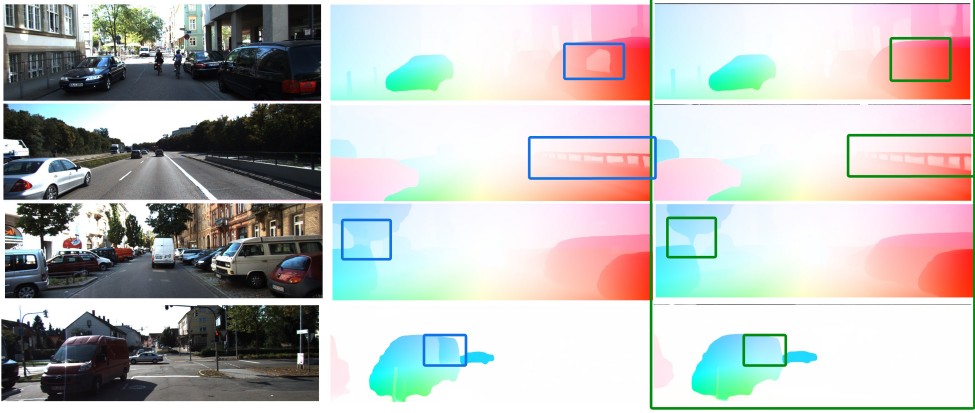

Figure 5: Qualitative results of optical flows: left column the original image, middle column is the optical flow of the original RAFT model, the right column(circled by green box) is the output of our model.

## 5  CONCLUSIONS

In this paper we have proposed a novel and effective semi-supervised learning strategy, with iterative pseudo label refinement and contrastive flow loss. Our framework aims at transferring the knowledge pretrained on the synthetic data to the target real domain. Through the iterative pseudo label refinement, we can leverage the ubiquitous, unlabeled real data to facilitate dense optical flow training and bridge the domain gap between the synthetic and the real. The contrastive flow loss is applied on a pair of corresponding features (one is warped to another via the pseudo ground truth flow), to boost accurate matching due to reliable pseudo labels and to dampen mismatching due to noisy pseudo labels, occlusion or global motion. Experiments results on KITTI 2015 and Sintel using two backbones RAFT and CRAFT, demonstrate the effectiveness of our proposed semi-supervised learning framework. We obtain the second best result (F1-all error of 4.11%) on KITTI 2015 benchmark among all the non-stereo methods (*c.f.* Sec. A for detailed rankings).

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

# A  APPENDIX

We show additional results in this appendix.

## A.1  SCREENSHOTS OF KITTI 2015 OPTICAL FLOW BENCHMARK

Fig. 6 shows the evaluation results on the KITTI 2015 optical flow benchmark [3]. Based on the baseline RAFT (Teed & Deng, 2020), our model RAFT-CF-PL3 (with CF for contrastive flow, and PL3 for iterative pseudo labeling at the 3rd iteration) obtains an F1-all error of 4.11%, *i.e.* a 19% error reduction with respect to RAFT (5.10%). Our model outperforms most of the evaluated non-stereo methods [4] as the time of submission, except for RAFT-OCTC (Jeong et al., 2022). The not-trivial improvement compared with RAFT, demonstrates the effectiveness of i) our proposed semi-supervised iterative pseudo labeling training strategy and, ii) the proposed contrastive flow loss, which facilitates the semi-supervised training by dampening the mismatching or displacement due to motion blur or occlusion when leveraging the pseudo labeling.

## A.2  SCREENSHOTS OF SINTEL BENCHMARK

Fig. 7 and Fig. 8 show the evaluation results on the MPI Sintel benchmark [5] on clean pass and final pass, respectively. Upon the baseline RAFT (Teed & Deng, 2020), our model RAFT-CF (with CF for contrastive flow, and no iterative pseudo labeling due to Sintel having ground truth labels) obtains an epe of 1.519 (clean pass), *i.e.* a 6% error reduction from RAFT with 1.609; and an epe of 2.645 (final pass), a 7% error reduction from RAFT with 2.855. The improved results show the effectiveness of our proposed contrastive flow loss. It helps the supervised training on the synthetic Sintel dataset (Butler et al., 2012a) by mitigating the mismatching due to occlusion, small fast-moving objects, and global motion, even though having the ground truth optical flow.

---

[3]www.cvlibs.net/datasets/kitti/eval_scene_flow.php?benchmark=flow

[4]Stereo methods uses left and right (stereo) images, but RAFT and ours use left images only.

[5]http://sintel.is.tue.mpg.de/quant?metric_id=0&selected_pass=0

Evaluation ground truth [All pixels ▾]    Evaluation area [All pixels ▾]

| | Method | Setting | Code | Fl-bg | Fl-fg | Fl-all | Density | Runtime | Environment | Compare |
|---|---|---|---|---|---|---|---|---|---|---|
| 1 | CamLiFlow++ | [icon] | | 2.07 % | 6.77 % | 2.85 % | 100.00 % | 1 s | GPU @ 2.5 Ghz (Python + C/C++) | ☐ |
| 2 | CamLiFlow | [icon] | code | 2.31 % | 7.04 % | 3.10 % | 100.00 % | 1.2 s | GPU @ 2.5 Ghz (Python + C/C++) | ☐ |

H. Liu, T. Lu, Y. Xu, J. Liu, W. Li and L. Chen: CamLiFlow: Bidirectional Camera-LiDAR Fusion for Joint Optical Flow and Scene Flow Estimation. CVPR 2022.

| | Method | Setting | Code | Fl-bg | Fl-fg | Fl-all | Density | Runtime | Environment | Compare |
|---|---|---|---|---|---|---|---|---|---|---|
| 3 | M-FUSE | [icon] | | 2.66 % | 7.47 % | 3.46 % | 100.00 % | 1.3 s | GPU | ☐ |
| 4 | RigidMask+ISF | | code | 2.63 % | 7.85 % | 3.50 % | 100.00 % | 3.3 s | GPU @ 2.5 Ghz (Python) | ☐ |

G. Yang and D. Ramanan: Learning to Segment Rigid Motions from Two Frames. CVPR 2021.

| | Method | Setting | Code | Fl-bg | Fl-fg | Fl-all | Density | Runtime | Environment | Compare |
|---|---|---|---|---|---|---|---|---|---|---|
| 5 | RAFT-OCTC | | | 3.72 % | 5.39 % | 4.00 % | 100.00 % | 0.2 s | GPU @ 2.5 Ghz (Python) | ☐ |

J. Jeong, J. Lin, F. Porikli and N. Kwak: Imposing Consistency for Optical Flow Estimation (Qualcomm AI Research). CVPR 2022.

| | Method | Setting | Code | Fl-bg | Fl-fg | Fl-all | Density | Runtime | Environment | Compare |
|---|---|---|---|---|---|---|---|---|---|---|
| 6 | SF2SE3 | [icon] | code | 3.17 % | 8.79 % | 4.11 % | 100.00 % | 2.7 s | GPU @ >3.5 Ghz (Python) | ☐ |

L. Sommer, P. Schröppel and T. Brox: SF2SE3: Clustering Scene Flow into SE (3)-Motions via Proposal and Selection. DAGM German Conference on Pattern Recognition 2022.

| | Method | Setting | Code | Fl-bg | Fl-fg | Fl-all | Density | Runtime | Environment | Compare |
|---|---|---|---|---|---|---|---|---|---|---|
| 7 | RAFT-CF-PL3 | | | 3.80 % | 5.65 % | 4.11 % | 100.00 % | 0.05 s | GPU @ 2.5 Ghz (Python) | ☐ |
| 8 | DIP | | | 3.86 % | 5.96 % | 4.21 % | 100.00 % | 0.15 s | 1 core @ 2.5 Ghz (Python) | ☐ |
| 9 | RAFT-3D | [icon] | | 3.39 % | 8.79 % | 4.29 % | 100.00 % | 2 s | GPU @ 2.5 Ghz (Python + C/C++) | ☐ |

Z. Teed and J. Deng: RAFT-3D: Scene Flow using Rigid-Motion Embeddings. arXiv preprint arXiv:2012.00726 2020.

| | Method | Setting | Code | Fl-bg | Fl-fg | Fl-all | Density | Runtime | Environment | Compare |
|---|---|---|---|---|---|---|---|---|---|---|
| 10 | RAFT-it | | | 4.11 % | 5.34 % | 4.31 % | 100.00 % | 0.1 s | GPU @ 2.5 Ghz (Python) | ☐ |
| 11 | GMFlow_RVC | | | 4.16 % | 5.67 % | 4.41 % | 100.00 % | 0.2 s | GPU @ 2.5 Ghz (Python) | ☐ |
| 12 | CCH-Flow | | | 4.20 % | 5.50 % | 4.42 % | 100.00 % | 0.2 s | 1 core @ 2.5 Ghz (Python) | ☐ |
| 13 | GMFlow+ | | | 4.27 % | 5.60 % | 4.49 % | 100.00 % | 0.2 s | GPU (Python) | ☐ |
| 14 | SeparableFlow | | code | 4.25 % | 5.92 % | 4.53 % | 100.00 % | 0.5 s | GPU | ☐ |

F. Zhang, O. Woodford, V. Prisacariu and P. Torr: Separable Flow: Learning Motion Cost Volumes for Optical Flow Estimation. Proceedings of the IEEE/CVF International Conference on Computer Vision 2021.

| | Method | Setting | Code | Fl-bg | Fl-fg | Fl-all | Density | Runtime | Environment | Compare |
|---|---|---|---|---|---|---|---|---|---|---|
| 15 | MetaFlow | | | 4.11 % | 6.77 % | 4.55 % | 100.00 % | 0.2 s | 1 core @ 2.5 Ghz (Python) | ☐ |
| 16 | KPA-Flow | | | 4.17 % | 6.77 % | 4.60 % | 100.00 % | 0.2 s | GPU @ 2.5 Ghz (Python) | ☐ |

A. Luo, F. Yang, X. Li and S. Liu: Learning Optical Flow With Kernel Patch Attention. Proceedings of the IEEE/CVF Conference on Computer Vision and Pattern Recognition 2022.

| | Method | Setting | Code | Fl-bg | Fl-fg | Fl-all | Density | Runtime | Environment | Compare |
|---|---|---|---|---|---|---|---|---|---|---|
| 17 | SwinTR-RAFT | | code | 4.32 % | 6.05 % | 4.61 % | 100.00 % | 0.6 s | 8 cores @ 2.5 Ghz (Python) | ☐ |
| 18 | RealFlow | | | 4.20 % | 6.76 % | 4.63 % | 100.00 % | 0.2 s | 8 cores @ 2.5 Ghz (Python) | ☐ |
| 19 | DGA-Flow | | | 4.34 % | 6.11 % | 4.64 % | 100.00 % | 0.2 s | 1 core @ 2.5 Ghz (Python) | ☐ |
| 20 | FCTR-m | | | 4.45 % | 5.63 % | 4.65 % | 100.00 % | 0.2 s | GPU @ 2.5 Ghz (Python) | ☐ |
| 21 | FlowNAS-RAFT-K | | | 4.36 % | 6.25 % | 4.67 % | 100.00 % | 0.19 s | GPU @ 2.5 Ghz (Python) | ☐ |
| 22 | FlowFormer | | code | 4.37 % | 6.18 % | 4.68 % | 100.00 % | 0.3 s | GPU (Python) | ☐ |

Z. Huang, X. Shi, C. Zhang, Q. Wang, K. Cheung, H. Qin, J. Dai and H. Li: FlowFormer: A Transformer Architecture for Optical Flow. European conference on computer vision 2022.

| | Method | Setting | Code | Fl-bg | Fl-fg | Fl-all | Density | Runtime | Environment | Compare |
|---|---|---|---|---|---|---|---|---|---|---|
| 23 | CRAFT-intramodes2 | | code | 4.35 % | 6.35 % | 4.68 % | 100.00 % | 0.2 s | 1 core @ 2.5 Ghz (Python) | ☐ |
| 24 | TPCV+RAFT | [icon] | | 4.53 % | 5.52 % | 4.69 % | 100.00 % | 0.2 s | 1 core 2.5ghz gpu | ☐ |
| 25 | UberATG-DRISF | [icon] | | 3.59 % | 10.40 % | 4.73 % | 100.00 % | 0.75 s | CPU+GPU @ 2.5 Ghz (Python) | ☐ |

W. Ma, S. Wang, R. Hu, Y. Xiong and R. Urtasun: Deep Rigid Instance Scene Flow. CVPR 2019.

| | Method | Setting | Code | Fl-bg | Fl-fg | Fl-all | Density | Runtime | Environment | Compare |
|---|---|---|---|---|---|---|---|---|---|---|
| 26 | AdaMatch | | code | 4.46 % | 6.23 % | 4.75 % | 100.00 % | 0.2 s | 1 core @ 2.5 Ghz (Python) | ☐ |
| 27 | Super | | | 4.43 % | 6.43 % | 4.76 % | 100.00 % | 0.07 s | GPU @ 2.5 Ghz (Python) | ☐ |
| 28 | RAFT-A | | code | 4.54 % | 5.99 % | 4.78 % | 100.00 % | 0.7 s | GPU @ 2.5 Ghz (Python + C/C++) | ☐ |

D. Sun, D. Vlasic, C. Herrmann, V. Jampani, M. Krainin, H. Chang, R. Zabih, W. Freeman and C. Liu: AutoFlow: Learning a Better Training Set for Optical Flow. CVPR 2021.

| | Method | Setting | Code | Fl-bg | Fl-fg | Fl-all | Density | Runtime | Environment | Compare |
|---|---|---|---|---|---|---|---|---|---|---|
| 29 | CRAFT | | code | 4.58 % | 5.85 % | 4.79 % | 100.00 % | 0.2 s | GPU @ 2.5 Ghz (Python) | ☐ |

X. Sui, S. Li, X. Geng, Y. Wu, X. Xu, Y. Liu, R. Goh and H. Zhu: CRAFT: Cross-Attentional Flow Transformers for Robust Optical Flow. CVPR 2022.

| | Method | Setting | Code | Fl-bg | Fl-fg | Fl-all | Density | Runtime | Environment | Compare |
|---|---|---|---|---|---|---|---|---|---|---|
| 30 | GMFlowNet | | code | 4.39 % | 6.84 % | 4.79 % | 100.00 % | 0.5 s | GPU @ 2.5 Ghz (Python) | ☐ |

S. Zhao, L. Zhao, Z. Zhang, E. Zhou and D. Metaxas: Global Matching with Overlapping Attention for Optical Flow Estimation. CVPR 2022.

| | Method | Setting | Code | Fl-bg | Fl-fg | Fl-all | Density | Runtime | Environment | Compare |
|---|---|---|---|---|---|---|---|---|---|---|
| 31 | CRAFT-autoflow | | | 4.50 % | 6.54 % | 4.84 % | 100.00 % | 0.1 s | GPU @ 2.5 Ghz (Python) | ☐ |
| 32 | SKFlow | | | 4.64 % | 5.83 % | 4.84 % | 100.00 % | 0.2 s | GPU @ 2.5 Ghz (Python) | ☐ |
| 33 | RAFT-DFlow | | | 4.52 % | 6.48 % | 4.84 % | 100.00 % | 0.2 s | 1 core @ 2.5 Ghz (C/C++) | ☐ |
| 34 | RAFT-FS | | | 4.60 % | 6.09 % | 4.85 % | 100.00 % | 0.15 s | GPU @ 2.5 Ghz (Python) | ☐ |

Figure 6: Screenshot of the KITTI 2015 optical flow benchmark Menze & Geiger (2015) evaluation results. Our model RAFT-CF-PL3 (CF for contrastive flow, and PL3 for iterative pseudo labeling at 3rd iteration) modified based on the backbone RAFT, obtains an F1-all error of 4.11%, *i.e.* a 19% error reduction from RAFT (5.10%), ranking the 2nd place among all evaluated non-stereo methods as of September 28, 2022. Our model is slightly outperformed by RAFT-OCTC (Jeong et al., 2022).

| | EPE all | EPE matched | EPE unmatched | d0-10 | d10-60 | d60-140 | s0-10 | s10-40 | s40+ | |
|---|---|---|---|---|---|---|---|---|---|---|
| GroundTruth [1] | 0.000 | 0.000 | 0.000 | 0.000 | 0.000 | 0.000 | 0.000 | 0.000 | 0.000 | Visualize Results |
| GMFlow+ [2] | 1.028 | 0.335 | 6.680 | 0.868 | 0.264 | 0.183 | 0.227 | 0.689 | 5.826 | Visualize Results |
| SplatFlow [3] | 1.119 | 0.511 | 6.061 | 1.410 | 0.394 | 0.247 | 0.272 | 0.868 | 5.915 | Visualize Results |
| FlowFormer [4] | 1.196 | 0.406 | 7.627 | 1.137 | 0.310 | 0.192 | 0.253 | 0.800 | 6.826 | Visualize Results |
| SKFlow [5] | 1.312 | 0.567 | 7.379 | 1.510 | 0.453 | 0.231 | 0.300 | 0.969 | 7.159 | Visualize Results |
| MS_RAFT [6] | 1.374 | 0.479 | 8.678 | 1.340 | 0.379 | 0.224 | 0.221 | 0.767 | 8.572 | Visualize Results |
| SwinTR-RAFT [7] | 1.379 | 0.529 | 8.304 | 1.272 | 0.430 | 0.277 | 0.325 | 0.917 | 7.726 | Visualize Results |
| GMA [8] | 1.388 | 0.582 | 7.963 | 1.537 | 0.461 | 0.278 | 0.331 | 0.963 | 7.662 | Visualize Results |
| GMFlowNet [9] | 1.390 | 0.520 | 8.486 | 1.275 | 0.395 | 0.293 | 0.314 | 0.991 | 7.698 | Visualize Results |
| GMA+LCT-Flow [10] | 1.408 | 0.525 | 8.611 | 1.428 | 0.404 | 0.251 | 0.279 | 0.876 | 8.299 | Visualize Results |
| AGF-Flow3 [11] | 1.409 | 0.525 | 8.618 | 1.433 | 0.403 | 0.250 | 0.278 | 0.878 | 8.303 | Visualize Results |
| RFPM [12] | 1.411 | 0.494 | 8.884 | 1.335 | 0.400 | 0.221 | 0.273 | 0.879 | 8.345 | Visualize Results |
| RAFT-OCTC [13] | 1.419 | 0.541 | 8.574 | 1.455 | 0.442 | 0.242 | 0.301 | 0.940 | 8.118 | Visualize Results |
| GMA-FS [14] | 1.430 | 0.602 | 8.171 | 1.579 | 0.470 | 0.263 | 0.333 | 0.977 | 7.961 | Visualize Results |
| AGFlow [15] | 1.431 | 0.559 | 8.541 | 1.501 | 0.452 | 0.261 | 0.319 | 0.963 | 8.075 | Visualize Results |
| DIP [16] | 1.435 | 0.519 | 8.919 | 1.102 | 0.407 | 0.312 | 0.336 | 0.754 | 8.546 | Visualize Results |
| CRAFT [17] | 1.441 | 0.611 | 8.204 | 1.574 | 0.552 | 0.249 | 0.311 | 0.991 | 8.131 | Visualize Results |
| GMA-base [18] | 1.450 | 0.591 | 8.440 | 1.532 | 0.470 | 0.280 | 0.321 | 0.951 | 8.251 | Visualize Results |
| SKFlow_RAFT [19] | 1.461 | 0.617 | 8.346 | 1.595 | 0.514 | 0.307 | 0.323 | 1.021 | 8.173 | Visualize Results |
| Anonymous [20] | 1.471 | 0.616 | 8.445 | 1.537 | 0.551 | 0.253 | 0.311 | 0.933 | 8.532 | Visualize Results |
| SeparableFlow-2views [21] | 1.496 | 0.567 | 9.075 | 1.474 | 0.481 | 0.257 | 0.309 | 0.958 | 8.691 | Visualize Results |
| DEQ-Flow-H [22] | 1.498 | 0.548 | 9.239 | 1.427 | 0.477 | 0.232 | 0.296 | 0.976 | 8.720 | Visualize Results |
| RAFT-CF [23] | 1.519 | 0.501 | 9.821 | 1.309 | 0.424 | 0.228 | 0.261 | 0.827 | 9.441 | Visualize Results |
| FCTR-m [24] | 1.524 | 0.575 | 9.264 | 1.512 | 0.468 | 0.250 | 0.325 | 0.979 | 8.791 | Visualize Results |
| RAFTwarm+AOIR [25] | 1.544 | 0.551 | 9.656 | 1.515 | 0.412 | 0.280 | 0.279 | 0.941 | 9.290 | Visualize Results |
| MFR [26] | 1.545 | 0.593 | 9.295 | 1.536 | 0.477 | 0.299 | 0.348 | 1.023 | 8.736 | Visualize Results |
| RAFT-it [27] | 1.554 | 0.612 | 9.242 | 1.664 | 0.514 | 0.273 | 0.287 | 0.971 | 9.261 | Visualize Results |
| SCAR [28] | 1.579 | 0.608 | 9.498 | 1.613 | 0.499 | 0.285 | 0.314 | 1.018 | 9.210 | Visualize Results |
| RAFTwarm+OBS [29] | 1.593 | 0.600 | 9.692 | 1.532 | 0.507 | 0.309 | 0.300 | 0.989 | 9.470 | Visualize Results |
| RAFTv2-OER-warm-start [30] | 1.594 | 0.625 | 9.487 | 1.567 | 0.512 | 0.339 | 0.328 | 1.014 | 9.271 | Visualize Results |
| AdaMatch [31] | 1.597 | 0.645 | 9.366 | 1.555 | 0.546 | 0.364 | 0.315 | 0.981 | 9.451 | Visualize Results |
| submission5367 [32] | 1.601 | 0.636 | 9.471 | 1.613 | 0.545 | 0.312 | 0.326 | 0.971 | 9.456 | Visualize Results |
| RAFT [33] | 1.609 | 0.623 | 9.647 | 1.621 | 0.518 | 0.301 | 0.341 | 1.036 | 9.288 | Visualize Results |

Figure 7: Screenshot of the MPI Sintel (clean pass) benchmark evaluation results. Our model RAFT-CF (CF for contrastive flow, and no iterative pseudo labeling due to Sintel having ground truth labels) modified based on the backbone RAFT, obtains an end-point-error (epe) of 1.519, *i.e.* a 6% error reduction from RAFT (1.609), showing the effectiveness of our proposed contrastive flow loss.

| | EPE all | EPE matched | EPE unmatched | d0-10 | d10-60 | d60-140 | s0-10 | s10-40 | s40+ | |
|---|---|---|---|---|---|---|---|---|---|---|
| GroundTruth [1] | 0.000 | 0.000 | 0.000 | 0.000 | 0.000 | 0.000 | 0.000 | 0.000 | 0.000 | Visualize Results |
| SplatFlow [2] | 2.072 | 1.063 | 10.285 | 2.717 | 0.851 | 0.452 | 0.508 | 1.538 | 11.109 | Visualize Results |
| FlowFormer [3] | 2.120 | 0.986 | 11.368 | 2.474 | 0.791 | 0.453 | 0.519 | 1.470 | 11.638 | Visualize Results |
| SKFlow [4] | 2.241 | 1.128 | 11.314 | 2.735 | 0.880 | 0.507 | 0.564 | 1.625 | 12.048 | Visualize Results |
| GMFlow+ [5] | 2.367 | 1.095 | 12.739 | 2.097 | 0.808 | 0.708 | 0.453 | 1.328 | 14.377 | Visualize Results |
| Anonymous [6] | 2.380 | 1.178 | 12.185 | 2.833 | 0.989 | 0.560 | 0.566 | 1.729 | 12.956 | Visualize Results |
| CRAFT [7] | 2.417 | 1.163 | 12.637 | 2.837 | 1.012 | 0.547 | 0.538 | 1.623 | 13.656 | Visualize Results |
| GMA-FS [8] | 2.441 | 1.203 | 12.551 | 2.777 | 0.961 | 0.594 | 0.587 | 1.646 | 13.576 | Visualize Results |
| AGFlow [9] | 2.469 | 1.221 | 12.643 | 2.892 | 0.991 | 0.698 | 0.560 | 1.692 | 13.816 | Visualize Results |
| GMA [10] | 2.470 | 1.241 | 12.501 | 2.863 | 1.057 | 0.653 | 0.566 | 1.817 | 13.492 | Visualize Results |
| RAFT-OCTC [11] | 2.574 | 1.243 | 13.435 | 2.880 | 1.045 | 0.667 | 0.578 | 1.701 | 14.594 | Visualize Results |
| SKFlow_RAFT [12] | 2.607 | 1.288 | 13.352 | 2.977 | 1.018 | 0.654 | 0.642 | 1.769 | 14.379 | Visualize Results |
| RAFT-CF [13] | 2.645 | 1.218 | 14.289 | 2.775 | 1.051 | 0.639 | 0.547 | 1.524 | 15.775 | Visualize Results |
| GMFlowNet [14] | 2.648 | 1.271 | 13.882 | 2.818 | 1.050 | 0.776 | 0.699 | 1.784 | 14.417 | Visualize Results |
| AGF-Flow [15] | 2.651 | 1.275 | 13.853 | 2.605 | 0.877 | 0.828 | 0.612 | 1.520 | 15.489 | Visualize Results |
| MS_RAFT [16] | 2.667 | 1.190 | 14.706 | 2.635 | 0.941 | 0.749 | 0.468 | 1.511 | 16.377 | Visualize Results |
| SeparableFlow-2views [17] | 2.667 | 1.275 | 14.013 | 2.937 | 1.056 | 0.620 | 0.580 | 1.738 | 15.269 | Visualize Results |
| ALNF [18] | 2.679 | 1.304 | 13.880 | 2.636 | 0.903 | 0.857 | 0.645 | 1.551 | 15.486 | Visualize Results |
| FCTR-m [19] | 2.687 | 1.261 | 14.310 | 2.897 | 1.032 | 0.709 | 0.578 | 1.761 | 15.386 | Visualize Results |
| RAFT+NCUP [20] | 2.692 | 1.323 | 13.854 | 3.139 | 1.086 | 0.636 | 0.635 | 1.844 | 14.949 | Visualize Results |
| AGF-Flow3 [21] | 2.733 | 1.217 | 15.105 | 2.419 | 0.912 | 0.737 | 0.463 | 1.440 | 17.133 | Visualize Results |
| GMA+LCT-Flow [22] | 2.734 | 1.218 | 15.103 | 2.419 | 0.914 | 0.738 | 0.465 | 1.441 | 17.131 | Visualize Results |
| submission5367 [23] | 2.742 | 1.282 | 14.656 | 3.027 | 1.110 | 0.644 | 0.562 | 1.743 | 15.980 | Visualize Results |
| L2L-Flow-ext-warm [24] | 2.780 | 1.319 | 14.697 | 3.098 | 1.145 | 0.637 | 0.656 | 1.879 | 15.502 | Visualize Results |
| LCT-Flow2 [25] | 2.781 | 1.349 | 14.465 | 2.720 | 0.989 | 0.895 | 0.620 | 1.582 | 16.405 | Visualize Results |
| RAFT-FS [26] | 2.785 | 1.341 | 14.557 | 3.114 | 1.104 | 0.649 | 0.681 | 1.850 | 15.487 | Visualize Results |
| MFR [27] | 2.801 | 1.380 | 14.385 | 3.075 | 1.112 | 0.772 | 0.674 | 1.829 | 15.703 | Visualize Results |
| RAFTwarm+AOIR [28] | 2.813 | 1.371 | 14.565 | 3.088 | 1.099 | 0.727 | 0.603 | 1.781 | 16.271 | Visualize Results |
| AdaMatch [29] | 2.817 | 1.331 | 14.937 | 3.008 | 1.119 | 0.773 | 0.661 | 1.805 | 15.969 | Visualize Results |
| NASFlow [30] | 2.822 | 1.403 | 14.389 | 2.998 | 1.146 | 0.910 | 0.655 | 1.757 | 16.143 | Visualize Results |
| RAFTwarm+OBS [31] | 2.826 | 1.356 | 14.809 | 3.134 | 1.116 | 0.735 | 0.631 | 1.832 | 16.117 | Visualize Results |
| RAFTv2-OER-warm-start [32] | 2.831 | 1.396 | 14.536 | 3.109 | 1.133 | 0.742 | 0.628 | 1.798 | 16.259 | Visualize Results |
| DIP [33] | 2.834 | 1.282 | 15.485 | 2.723 | 1.090 | 0.801 | 0.571 | 1.812 | 16.531 | Visualize Results |
| DEQ-Flow-H [34] | 2.851 | 1.366 | 14.956 | 2.984 | 1.211 | 0.764 | 0.554 | 1.840 | 16.688 | Visualize Results |
| raft-jm [35] | 2.854 | 1.365 | 14.983 | 3.089 | 1.102 | 0.796 | 0.650 | 1.769 | 16.403 | Visualize Results |
| RAFT [36] | 2.855 | 1.405 | 14.680 | 3.112 | 1.133 | 0.770 | 0.634 | 1.823 | 16.371 | Visualize Results |

Figure 8: Screenshot of the MPI Sintel (final pass) benchmark evaluation results. Our model RAFT-CF (CF for contrastive flow, and no iterative pseudo labeling due to Sintel having ground truth labels) modified based on the backbone RAFT, obtains an end-point-error (epe) of 2.645, *i.e.* a 7% error reduction from RAFT (2.855), showing the effectiveness of our proposed contrastive flow loss.

