# OpenReview forum: "CLIP-FLOW: CONTRASTIVE LEARNING WITH ITERATIVE PSEUDO LABELING FOR OPTICAL FLOW"
_ICLR.cc/2023/Conference — Submitted to ICLR 2023_

### Official Review · Reviewer_bSsw · 2022-10-22

**Confidence:** 3
**Correctness:** 3
**Technical Novelty And Significance:** 2
**Empirical Novelty And Significance:** 2
**Recommendation:** 5

**Clarity, Quality, Novelty And Reproducibility:**

Clarity: Good. This paper is well-written and easy to understand.
Quality: Fair. See the weakness section - some of the ablations are missing.
Novelty: Good. As far as I know, the proposed techniques are not seen in previous optical flow research.
Reproducibility: Unknown.

**Strength And Weaknesses:**

Strength:
- The overall writing is clear and easy to understand.
- The related work section covers quite a complete set of recent works on optical flow, semi-supervised and representation learning.
- The proposed method is simple and achieves good results in a semi-supervised setting.

Weaknesses:
- The iterative algorithm is not quite clear:
  - For input $\phi_\text{our}^0$, should it be "Ours ( $\phi_\text{bs}$+ pseudo labeling + contrastive flow loss), ..."?
  - Why the 0-th round pseudo-labels are generated by $\phi_\text{bs}$ rather than $\phi_\text{our}^0$? Since the assumption is that $\phi_\text{our}^0$ is better than $\phi_\text{bs}$?
- "shows significant improvement for synthetic dataset as well" - it seems I could not find relevant experiments in the paper to back this up.
- The ablation is not quite complete: there are three orthogonal parts in the proposed method, and CE and CF can be applied to RAFT and CRAFT without the pseudo-label re-training part. Wondering how the performances are for RAFT-CE-CF and CRAFT-CE-CF on KT15 (without PL)?
  - And for the ablation experiments in table 3 and figure 4, it would be nice to show the performance after conducting the same 3-round iterative training process.
  - Is the fine-tuning step on $D_R^{\text{tr}}$ necessary?
  - How to choose the $S$ for the total training steps on $D_R^{\text{u}}$?
- Limitations are not discussed in the paper.
- Will the authors release code upon acceptance?

Minor:
- missing bracket on the line right above 3.3: ($\mathcal{D}_R$ is marginal
- in section 4.2, "we finetune the updated RAFT-CF-CE-PL model from part A": should this be RAFT-CF-CE?

**Summary Of The Paper:**

This paper proposes CLIP-FLOW, which is a method to improve the performance of optical flow models under the semi-supervised learning setting. The proposed method consists of three orthogonal parts: 1) Semi-supervised Contrastive Flow: an additional contrastive loss for training optical flow models; 2) Coordinate Encoding: an additional 2D-coordinate map of each pixel as two additional channels to the original input frames; 3) Semi-supervised iterative training on unlabeled real data by pseudo-labeling. They show on KITTI 2015 dataset the proposed process can improve RAFT and CRAFT models, and achieve a second place on the benchmark.

**Summary Of The Review:**

Overall I think the proposed method makes sense and the performance on KITTI 2015 is good. But I am mostly concerned about the limited ablation of the proposed model.

----
After rebuttal comments:
Thanks the authors for carrying out the rebuttal. But my concerns on limited ablation are not fully addressed:
1) I think RAFT-CE-CF and CRAFT-CE-CF evaluation is still necessary for validating the effectiveness of CE-CF alone.
2) The necessity of fine-tuning step on $D_R^{\text{tr}}$ should be validated.
and for "shows significant improvement for the synthetic dataset as well", I was wondering if the final model, rather than only RAFT-CF-CE, also has significant improvement for the synthetic dataset?

---

> ### Author Response · Authors · 2022-11-19
> **Response to the reviewer bSsw**
>
> ### Comment 1: Clarification in Algorithm
> ${\phi_{our}^0}$ is rightly mentioned as ours in the algorithm, as this is the initial model, which has the same model architecture as $\phi_{bs}$, but also has contrastive loss and pseudo labeling modules added to it. We select $\phi_{bs}$ instead of ${\phi_{our}^0}$ to generate pseudo labels only for the first iteration because, during the first iteration, ${\phi_{our}^0}$ denotes the model, which is still not trained with contrastive loss and pseudo-labeling.
>
> ___
>
> ### Comment 2: Relevant experiments for improvement on Synthetic dataset
> As shown in Table.1 (main paper), Ours (RAFT-CF-CE) on Sinel outperforms the original RAFT. To further address reviewers’ concerns, we conduct the same experiments on Sintel and present the results in Table. 4.
>
> ___
>
> ### Comment 3: Wondering how the performances are for RAFT-CE-CF and CRAFT-CE-CF on KT15 (without PL)?
> Thanks for the suggestion. We understand that additional ablation studies help us to understand the model better in a different aspect. However, the suggested experiment aims to show the effectiveness of the PL module, which we have already shown in RAFT-PL v.s. RAFT. Please note that RAFT achieves 5.1 F1-all on KITTI-test while RAFT-PL is 4.48, which has already proven the effectiveness of the PL module.
>
> ___
>
> ### Comment 4: Limitations are not discussed in the paper.
> Our method is useful when the ground truth labels are very expensive to achieve. Our pseudo-labeling semi-supervised learning strategy is proved very effective. However, when dealing with large-scale datasets where ground truth labels are cheap to achieve, such as synthetic datasets, our method will not show significant improvement.
> ___
>
> ### Comment 5: Is fine-tuning step $D_R^{tr}$ necessary?
> Yes, finetuning the data with real GT labels is necessary.
> As we use baseline models to obtain pseudo labels, our method's performance is bounded by the baseline models without the finetuning step.
> ___
>
> ###  Comment 6: How to choose the  for the total training steps on $D_R^{u}$?
> We use a fixed the training steps for all the iteration for simplicity and comparison. In practice, the training steps can be setup according to the validation dataset.
> ___
>
> ### Comment 7: Will the authors release code upon acceptance?
> Yes, we are glad to release all the related codes upon acceptance.
> We believe our paper is self-contained and has included the details for re-implementation.

---

### Official Review · Reviewer_8YE1 · 2022-10-23

**Confidence:** 4
**Correctness:** 4
**Technical Novelty And Significance:** 1
**Empirical Novelty And Significance:** 2
**Recommendation:** 5

**Clarity, Quality, Novelty And Reproducibility:**

About the novelty of the paper:
1. The semi-supervised iterative labeling training strategy (also algorithm 1) seems to over-complicate the approach. I would suggest to compare with a simple baseline, which just trains on all pseudo-lablled data without k-Fold Cross validation (without iterations but with the same amount of epochs). Due to the iterations and model choosing after each iteration, the k-Fold Cross validation setting actually trains on all data, and I would like to see the performance difference from the simple baseline.
2. The contrastive loss is more like an auxiliary loss, and it has nothing to do with self-supervision. And thus, in my option, it would be better to design the contrastive loss such that the model can be pre-trained on a large number of unlabeled data. For example, we can use video data of consecutive frames and self-supervised losses (like intensity consistency), and see how contrastive loss can be fit into this setting.

Clarification:
1. Any insight on the 2D positional encoding? Wouldn't it confuse the model if we consider optical flow as a feature matching problem?
2. In section 3.1, RAFT "F1-all errors of 4.11 (ours) vs 5.10 (raft) on KITTI-2015" and CRAFT "F1-all errors of 4.66 (ours) vs 4.79 (craft) on KITTI-2015". Why CRAFT+this paper is worse thant RAFT+this paper?
3. In Table 2 and Table 3, why are some results are missing?
4. In Figure 4 and Figure 5, should the 3rd iteration and RAFT-CF-CE-PL match with each other?

The manuscript needs a proof-reading, as I see some obvious typos:
1. In abstract, line2, "," is not needed.
2. In introduction, paragraph 2, line 10, ", however" can be deleted.

**Strength And Weaknesses:**

Strength:

This work is a good attempt to explore the (synthetic and real) data usage given the difficulty of the optical flow labeling, and the experiments show good insight on the performance improvements we can obtain by pseudo labeling and contrastive loss. And I think the 2D positional encoding idea is interesting.

Weaknesses:

The technique novelty of this paper is not very strong. Pseudo labeling and contrastive loss have been widely used recently, and the design of the semi-supervised iterative labeling training strategy (also algorithm 1) seems to over-complicate the approach. See more details in the below section.

The writing can be improved as I see some obvious typos in the paper. See some examples in the below section.


**Summary Of The Paper:**

Due to the difficulty of labeling GT optical flow data, this paper aims to explore how to better use of synthetic data and transfer the knowledge from the synthetic domain to the real domain. Thus, the two main contributions of this paper are 1) the iterative pseudo labeling and 2) supervised contrastive flow loss. In addition, this paper also discusses the impact of 2D positional encoding in the optical flow task.

**Summary Of The Review:**

This work is a good attempt to explore the data usage given the difficulty of the optical flow, but its technique novelty is not very strong.

---

> ### Author Response · Authors · 2022-11-19
> **Thank you for comments and suggestions**
>
> ### Comment 1: Novelty is not strong
>
>  We humbly disagree with the reviewer. The paper is the first of its kind, which leverages synthetic datasets. It effectively combines contrastive loss and iterative pseudo refinement to achieve the state-of-art result for a real dataset, which lacks ground-truth labels. Our proposed method allows a pre-trained optical flow model to further improve itself in a positive feedback loop without any additional annotations or parameters. Albeit simple, our method beats all existing(published) methods, that in our view should be considered a strength rather than a weakness.
>
> ___
>
> ### Comment 2: Overcomplicated cross-validation strategy
>
> We understand the reviewer’s
> concern.  Given the fact that (1) we train the model for a larger number of iterations, (2) the KITTI 2015 Flow dataset has only 200 images with ground truth labels, it becomes imperative to avoid overfitting during the fine-tuning stage.  For this, we adopt an unbiased 5-fold cross-validation strategy, which helps us identify the optimum number of fine-tuning training steps. The validation process is quite straightforward and commonly used in various machine learning tasks. The effectiveness of this is well illustrated in Figure 3 in the main paper.
>
> ___
>
> ### Comment 3: Any insight on the 2D positional encoding?
>
>  Our main idea behind adding positional encoding was to add a spatial context, along with forcing the model to be more accurate when used in tandem with contrastive loss. Through our ablation experiments, we do find that 2D positional encoding helps improve the  accuracy consistently with or without contrastive loss.
>
> ___
>
> ### Comment 4: Why CRAFT+this paper is worse than RAFT+this paper?
>
> This can be largely attributed to the fact that CRAFT has a longer training time when compared to the RAFT-based model. In our experiments, we found the  CRAFT model is about 4 to 5 times slower than its RAFT counterpart. Given
> the current timeline, we were not in a position to run the CRAFT model for as many training steps as we ran the RAFT model. However, we should not fail to see the fact that we do see an incremental improvement of about  4.4%  for
> CRAFT when it is further trained with our contributions, see Table  1).
>
> ___
>
> ### Comment 5: In Table 2 and Table 3, why are some results are missing?
> Due to the limited number of submissions (3 per team) allowed per month, we cannot evaluate all of our models on the KITTI benchmark.  However, validation F1  results give a good and congruent idea about the probable performance of the model on the test dataset.
>
> ___
>
> ### Comment 6: In Figure 4 and Figure 5, should the 3rd iteration and RAFT-CF-  CE-PL match with each other?
>
> I believe you are talking about the curves in Figure 4(a) and Figure 4(b). first iteration curve (blue) in Figure 4(a) and
> RAFT-CF-CE-PL curve (blue) in Figure 4(b) refers to the same experiment.  Note that the 3rd iteration does not match any curve in Figure 4(b). We use the 1st iteration for the ablation study.

---

### Official Review · Reviewer_9251 · 2022-10-23

**Confidence:** 4
**Correctness:** 3
**Technical Novelty And Significance:** 2
**Empirical Novelty And Significance:** 2
**Recommendation:** 5

**Clarity, Quality, Novelty And Reproducibility:**

- Clarity: needs improvement.

  I think the paper needs improvement in clarity. For example, there are missing explanations on 'valid pixels', the final loss on page 6. Furthermore, it would be good to include the ablation study on the Sintel dataset too.

- Quality / Novelty: A bit limited

  If the paper can show that the idea generalizes to another dataset, e.g, Sintel, the paper would have more novelty. The paper included the accuracy on the Sintel test benchmark, but it's not so clear which factor contributes to the accuracy improvement on the Sintel dataset.

- Reproducibility: reproducible

  The paper provides enough details for reproducing the results. But providing source codes will be always helpful.

- By the way, **the paper title on this openreview page doesn't match** with the title in the pdf.

**Strength And Weaknesses:**

__Strength__

- Interesting idea: iterative training + pseudo ground truth generation

  The proposed semi-supervised learning scheme is interesting and is successfully validated through experiments. It was interesting to see that the accuracy on the validation set keeps improving along with the iteration steps, like a positive feedback loop.

- Contrastive learning loss

  The paper demonstrates the effectiveness of the contrastive learning loss for the optical flow task; it helps to learn more discriminative features for the matching task. Its contribution would affect and help other following-up works.

---

__Weakness__

- Limited generalization: Analysis on only one dataset with one model.

  The paper provides an ablation study only on the KITTI dataset. I wonder if the method can also show the same consistent accuracy improvement on the other dataset, such as Sintel. It would be great if the paper can demonstrate the same ablation study (e.g., Table 2 and 3) on the Sintel dataset. In my humble opinion, KITTI is a quite restricted setup with street scenes, perspective motion, and objects with rigid motion. Thus overfitting on the KITTI domain certainly improves the accuracy. On the other hand, Sintel shows much more diverse scenes with various visual effects.

   Furthermore, the paper shows the analysis with RAFT only. Can this scheme generalize to other models, such as CRAFT? Considering the accuracy improvement on CRAFT, it seems a bit limited. I wonder if the paper can also provide the same ablation study for CRAFT.

- Accuracy after the training with pseudo ground truth (i.e., before finetuning)

  I am wondering how the accuracy is, just right after the training with pseudo ground truth for each iteration step. (I guess that Table 2 shows the accuracy after the finetuning step.) I wonder if the accuracy after the pseudo ground truth training improves along with the number of iteration steps. it would be great to include the numbers in Table 2 as a separate column.

- Overlap between the unlabeled data and the validation set

  The KITTI Optical Flow 2015 dataset (which has ground truth labels) is actually included in the KITTI raw dataset. I am wondering if those images are excluded when during the training with pseudo ground truth.

- Contrastive loss on occluded region

   I wonder if the contrastive loss is also applied on an occluded region (or what are the "valid pixels" in the text on the page 6?). The contrastive loss shows its benefits but might hurt the accuracy when the loss is applied on occluded pixels because their corresponding pixels don't exist, so the loss will be applied to mismatching pixels.

- Final loss

  What is the final loss applied to the model? Is it a weighted sum of $L_q$ and $L_{CT}$? (eq.1 and eq. 3?)


**Summary Of The Paper:**

The paper introduces a semi-supervised optical flow approach based on an iterative pseudo-labeling scheme.
Given a pre-trained model (e.g., trained on synthetic datasets), the method computes a pseudo ground truth on an unlabeled target dataset and trains the model on it. The method then fine-tunes the model using a small amount of available ground truth data.
After that, the model generates the pseudo ground truth on the unlabeled target dataset again and loops the process.
The paper also proposes a contrastive loss that further improves the accuracy.
The proposed ideas consistently improve the accuracy of baseline models.

**Summary Of The Review:**

The proposed idea is interesting, and the experiments validate its effectiveness. However, the experiment is limited to only one dataset with one model. It would be really great to see more ablation studies using other models (e.g., CRAFT) on other datasets (e.g, Sintel) in order to see that the method can be generalized to other methods/datasets.

Furthermore, it would be great if some concerns about technical designs and experiment setups, as stated in the weakness section.


----

__After the authors' response__

I appreciate the authors' response! The responses resolved most of my main concerns. However, due to the following reasons, I would like to keep my original rating, marginally below the threshold.
- Is the contrastive loss really helpful? Numbers in the tables show that it doesn't give consistent improvement.
  - In the last two rows in Table 3 (main paper), adding the contrastive loss (CF) marginally degrades the test accuracy on KITTI 2015 test.
  - In the ablation study on Sintel (Table 4 in the authors' response), the accuracy dropped quite a bit on EPE-Final (test).
- Novelty concern: As other reviewers mentioned, I am not so sure the novelty is significant enough. It was a bit difficult to find exciting insights. That said, I don't strongly object to accepting the paper.
- If the target scenario is a real-world domain, an experiment with other real-world datasets would be also interesting (I am sorry that I didn't think about suggesting it before). It could be HD1k, Slow Flow (their ground truth might not be super accurate), Middlebury (not so many training images though), DAVIS (probably reporting keypoint tracking accuracy?), or any suitable dataset.

---

> ### Author Response · Authors · 2022-11-19
> **Response to reviewer 9251**
>
> ## Reviewer R1(9251)
>
> Thanks for the detailed comments. We appreciate the suggestions and respond to them below.
>
> ___
>
>
> | iteration | steps | F1-all (val) | F1-all (test) |
> | ------------- | ------------- | ------------- |------------- |
> | CRAFT |  |  | 4.79
> | 1st | 620k | 5.57 | 4.66
> | 2nd | 100k | 5.26 | 4.58
>
> Table 1: The comparison of F1-all error on KITTI-Flow-Val and KITTI-Flow-test at different iterations and different training steps using CRAFT backbone.
>
> ___
>
>
>
> | Models (2nd iteration @ step 100k) | F1-all (validation) |
> | ------------- | ------------- |
> | CRAFT-PL | 5.37 |
> | CRAFT-CE-PL | 5.26 |
> | CRAFT-CF-CE-PL | 5.19 |
>
>
> Table 2: The comparison of F1-all error KITTI-Flow-Val and KITTI-Flow-test for different CRAFT-based models.
> ___
>
> ### Comment 1: Limited generalization: Analysis on one dataset.
>
> As outlined in the motivation of our work, we primarily focus on improving optical flow estimation for real dataset, which generally lacks ground-truth labels, using synthetic datasets in a semi-supervised fashion. KITTI is the prime real dataset, which has been regularly used in all previous works for quantitative and qualitative evaluation, so we have used KITTI.
>
> Having said that, we agree with the reviewer on the larger principle of generalization and have gone ahead and conducted a few carefully crafted ablation experiments for the Sintel dataset. In order to prove our method’s efficacy on synthetic datasets as well, we conduct a set of experiments strictly following our settings used for the KITTI dataset. We found that the Sintel dataset has a very diverse difficulty across different scenes. See the evaluation of on Sintel training set using RAFT (chairs + things) in Table. 3. Therefore, we divided the original training set into three splits (Sintel-train, Sintel-val, and Sintel-test) by including a variety of difficulty levels in each split.
>
> We present an additional ablation study in Table. 4. Baseline: We finetune the RAFT model trained with Flying Chairs and Flying Things (RAFT-CT) on Sintel-val as our Baseline model (RAFT-CTS). The best model is chosen to test on Sintel-test. Ours: We obtain pseudo labels of Sintel-train using RAFT-CTS and thus carry out the ablation studies. The result shows an improvement over Baseline with the use of our method. For RAFT-PL, we use the Baseline model to obtain the pseudo labels of the Sintel-train, which is thus used to supervise the training of the original RAFT model. Here we train the model for 40k steps, finetune it on the Sintel-val, and test the model on the Sintel-test. RAFT-CE-PL adds the coordinate encoding, and RAFT-CF-CE-PL adds the coordinates encoding and the contrastive flow loss.
>
> ___
>
> ### Comment 2: Limited generalization: Analysis on only one model
> Whereas we concur that extensive ablation using our training strategies is only done for the RAFT-based base model, we did verify the efficacy of our proposed training strategy even for CRAFT model. Our finding was RAFT-based model converges faster and beats the existing state-of-art result. In order to address the reviewer’s concern, we conducted ablation experiments with CRAFT as the base model. The results are shown in Table.1 and Table.2. As shown in Table.1, the iterative training strategy improves the CRAFT model consistently. Table.2 shows that our proposed modules are beneficial, regardless of the backbone model.
>
> ___
>
> ### Comment 3: Overlap between the unlabeled data and the validation set
>
> We do not remove the mentioned data points from the KITTI-Raw dataset. The rationale being, during the pseudo-label generation process, we do not use the GT labels for loss calculation, and during the fine-tuning stage, we precisely perform un-biased k-fold cross-validation to find the optimum number of training steps to avoid any possibility of overfitting, which is rightly validated by our test-results on KITTI-Test Flow leaderboard.
>
> ___
>
> ### Comment 4: Accuracy after the training with pseudo ground truth
>
> In fact, the F1-all (Val) result, presented in Table 2 (main paper), is the accuracy after the training with pseudo ground truth and without fine-tuning. We can see consistent improvements with the iterative training scheme.
>
> ___
>
> ### Comment 5: Final loss
>
> Eq.1 in the main paper is the traditional contrastive loss equation, and Eq.2 explains how we use the traditional contrastive loss in our optical flow formulation. Eq.3 is the final formulation of the contrastive flow loss we use in our method. Our final loss is the weighted-sum loss of the original RAFT loss (l1 loss) and contrastive flow loss(Eq. 3). Thank you for pointing out this. We will clarify this in our final version.
>
> ___

---

> > ### Author Response · Authors · 2022-11-19
> > **Response to Reviewer 9251**
> >
> > | Scene  |  EPE(Clean)  | EPE(Final)  |  Total Images  |
> > |---|---|---|---|
> > | alley$_2$ | 0.172395 | 0.198065 |  49 |
> > | ambush$_5$ |  2.321991 |  6.003811 |  49 |
> > | bamboo$_2$ | 0.604761 |  0.801916 |  49 |
> >   |  bandage$_1$ |  0.354612 |  0.421356 |  49 |
> >   |  bandage$_2$ |  0.240449 |  0.456826 |  49 |
> >   |  cave$_2$ | 3.564749 |  6.194697  |  49 |
> >   |    market$_2$ |  0.412411 |  0.645884 |  49 |
> >   |   market$_5$ |  4.776794 |  8.245067 |  49 |
> >   |    sleeping$_1$ | 0.114113 | 0.122474 | 49 |
> >   | sleeping$_2$ | 0.110270 | 0.113994 | 49  |
> >    |  ambush$_7$ | 0.267409 | 0.435432 | 49 |
> >    | mountain$_1$ | 0.198944 | 0.277270 | 49 |
> >    |  **shaman$_2$** | 0.214891 | 0.247883 | 49 |
> >   |  **temple$_3$** | 2.516815 | 4.225691 | 49 |
> >   | **cave$_4$** |  2.126021 | 2.811051 | 49 |
> >   | **ambush$_4$** |  9.265260 | 18.043339 | 32 |
> >    | **alley$_1$** | 0.183764 | 0.215643 |  49 |
> > | *shaman$_3$* | 0.163714 | 0.249726 | 49 |
> >    | *bamboo$_1$* | 0.418640 | 0.398496 | 49 |
> > | *temple$_2$* | 1.617814 | 2.373946 | 49 |
> >   | *ambush$_6$* | 3.235351 | 7.418845 | 19|
> >    | *market$_6$* | 1.963488 | 2.501761 | 36 |
> >   | *ambush$_2$* | 4.528018 | 19.891224 | 20 |
> >
> > Table 3: The evaluation result of RAFT (Chairs+Things) on the Sintel training set.
> > The **bold scenes** are used as our validation set, and the *italics scenes* are used as our test set. We use the remaining scenes plus the original Sintel test dataset as our training set in the experiment of pseudo-labeling training. Note that no ground truth labels are used in this pseudo-labeling experiment.

---

> > ### Author Response · Authors · 2022-11-19
> > **Response to Reviewer 9251**
> >
> >
> > | Method |  EPE-Clean(test)  | EPE-Final (test) |
> > |---:|:---:|:---:|
> > |Baseline |  2.193938 | 3.644036 |
> > |RAFT-PL | 1.877319 | 3.218146 |
> > |RAFT-CE-PL | 2.143940 |  3.116621  |
> > | RAFT-CF-CE-PL | 2.061207  |3.419203 |
> >
> > Table 4: Ablation study on the Sintel dataset

---

### Decision · Program_Chairs · 2023-01-20

**Decision:**

Reject

**Justification For Why Not Higher Score:**

All the main ideas in the paper, including semi-supervised learning using an iterative labeling scheme and contrastive learning for better features, are known recipes for ML models. There are no "fresh" ideas and "surprising" results.

**Justification For Why Not Lower Score:**

N/A

**Metareview: Summary, Strengths And Weaknesses:**

The paper introduced a semi-supervised optical flow approach based on an iterative labeling scheme on an unlabeled target dataset. A contrastive loss was introduced to improve the accuracy. The reviewers recognized the strength of the paper: the proposed semi-supervised learning scheme (iterative training + pseudo GT generation) is simple & interesting as a good attempt, and is successfully validated through the experiments; the paper demonstrated the effectiveness of contrastive learning loss of learning more discriminative features for optical flow. However, all the reviewers showed the concerns that the novelty and quality of the paper are limited. The contrastive loss is more like an auxiliary loss and has nothing to do with self-supervision. The whole semi-supervised iterative labeling training strategy is over-complicated. There were also concerns of limited ablation of the proposed method.

In the rebuttal, the authors added ablation studies on Sintel and results on CRAFT and clarified some questions. During the discussion, Reviewer 9251 felt the contrastive loss doesn’t generate consistent improvement, and an experiment with real-world datasets is important to justify the motivation of the paper. Reviewer bSsw still had concerns on limited ablation. The AC agrees with the reviewers that the novelty is limited as the paper puts many known ideas for a specific field of optical flow and showed somewhat limited improvement.